# Two-Dimensional Metal–Organic Framework TM Catalysts for Electrocatalytic $N_2$ and $CO_2$ Reduction: A Density Functional Theory Investigation

**Anqi She [1], Ming Wang [1], Shuang Li [1], Yanhua Dong [1,\*] and Dandan Wang [2,\*]**

[1]   Department of Mathematics and Computer Science, Jilin Normal University, Siping 136000, China;
      a15526580128@jlnu.edu.cn (A.S.); wm@jlnu.edu.cn (M.W.); 0412201-ls@163.com (S.L.)
[2]   Key Laboratory of Functional Materials Physics and Chemistry of Ministry of Education, College of Physics, Jilin Normal University, Changchun 130103, China
\*    Correspondence: computerdyp@jlnu.edu.cn (Y.D.); mila880227@126.com (D.W.)

**Abstract:** In this study, we screened novel two-dimensional metal–organic framework (MOF) materials, which can be used as efficient electrocatalysts in the $N_2$ reduction reaction (NRR) and $CO_2$ reduction reaction ($CO_2$RR) through density functional theory (DFT) calculations. By systematically investigating the adsorption behaviors of $N_2$ and $CO_2$ in different MOF-TMs (TM = Fe, Co, Ni, Cu, Zn) and their electrocatalytic hydrogenation processes, we found that 2D MOF-Fe, MOF-Co, and MOF-Ni can be used as catalysts for electrocatalytic NRR. The free energy increase in the corresponding potential-limiting step is calculated to be 0.84 eV on MOF-Fe, 1.00 eV on MOF-Co, and 1.17 eV on MOF-Ni, all of which are less than or at least comparable to those reported values for the NRR. Moreover, only 2D MOF-Fe was identified as a suitable electrocatalyst for $CO_2$RR. Instead of other hydrocarbons, the product $CH_3OH$ is selectively obtained in an electrocatalytic $CO_2$ reduction reaction on a 2D MOF-Fe with a free energy increase of 0.84 eV in the potential-limiting step. Overall, the results of this study not only facilitate the potential application of 2D MOF-TMs as electrocatalysts but also provide new guidelines for rationally designing novel electrocatalysts for the NRR and $CO_2$RR.

**Keywords:** metal–organic frameworks; DFT calculations; electrocatalyst; nitrogen reduction reactions; carbon dioxide reduction reactions

## 1. Introduction

In recent years, in order to alleviate the dual pressure on the environment and energy, nitrogen fixation and carbon dioxide reduction have become the focus of research. Among the various catalytic reduction methods for $N_2$ and $CO_2$ molecules, an electrocatalytic reduction reaction is considered as one of the most advantageous technologies [1–5]. For this reason, the development of new effective electrocatalysts for nitrogen reduction reactions (NRR) and carbon dioxide reduction reactions ($CO_2$RR) is urgently required. Efforts have been made to explore efficient electrocatalysts, and a significant overlap between the catalysts used for NRRs and $CO_2$RRs has been found. For example, noble metal catalysts, such as Au, [6–8] Pt, [9–11] Rh, [10,12], and Pd [8,10], are efficient NRR and $CO_2$RR electrocatalysts due to their electronic properties. Moreover, transition metal oxides, such as $TiO_2$, [13–15] $WO_{3-x}$, [16,17], $MnO_2$ [18–20], and CuO [21,22], are considered promising electrocatalysts for the NRR and $CO_2$RR according to their adjustable electronic structures and low cost. However, despite the development of noble metal and metal oxide electrocatalysts, they are limited by several problems, such as the high price of noble metals and the instability of metal oxides at negative potentials. Therefore, there is a broader interest in developing novel NRR and $CO_2$RR electrocatalysts with a low cost, high efficiency, and high stability.

Metal–organic frameworks (MOFs), which are one-, two-, or three-dimensional skeletons formed by metal atoms and organic bridging ligands through coordination bonds, are attracting increasing attention in electrocatalysis due to their large specific surface area, high stability, low cost, homogeneous active composition, and dense catalytic sites that are easy to tune. For example, Co-MOF, Fe-MOF, and Cu-MOF have been widely used as electrocatalysts for oxygen evolution reactions since Yaghi et al. reported the first MOFs in 1995 [23–26]. In particular, MOFs including Al-MOFs, Cu-MOFs, and Co-MOFs show great promise and have recently attracted extensive attention regarding applications in electrocatalytic NRRs and $CO_2$RRs [27–30]. However, MOFs with a three-dimensional morphology have several weak points, such as poor electrical conductivity, low metal utilization, and accessible surface area, which limit their electrocatalytic applications. To bypass these problems, two-dimensional (2D) MOFs with metallic property, unsaturated metal sites, and large surface areas have been investigated and show excellent performance in electrocatalysis [31,32]. Until now, the study of 2D MOFs has mainly focused on synthetic methods and their applications in energy storage [33,34]. There are limited reports on 2D MOFs for both electrocatalytic NRRs and $CO_2$RRs [35–37]. Therefore, it is of great significance to explore the potential applications of 2D MOFs in electrocatalytic NRRs/$CO_2$RRs and evaluate the corresponding NRR/$CO_2$RR performance.

Thus far, many metallic 2D MOFs that contain transition metal (TM) ions (Fe, Co, Ni, Mn, Mo) have been successfully synthesized and studied [35,36,38]. Over the last year, we have designed a 2D ultrathin MOF-Co for highly safe and long-life Li-S batteries [34]. The 2D MOF-Co possesses a unique structure, namely, periodically arranged cobalt atoms coordinated with oxygen atoms causing pseudo-octahedra, which are mutually interconnected by the 1,4-benzenedi-carboxylic acid ligands. Interestingly, the Co sites in Co-$O_4$ moieties are exposed on the surface. It is expected that the exposed coordination, with unsaturated metal sites on the 2D-MOF surface, can serve as active centers to ensure the high electrocatalytic activity for NRRs and $CO_2$RRs. Generally, metallic 2D MOFs have a high tunability because transition metals are diverse. Inspired by the above concepts, we studied the electrocatalytic performance of several 2D MOF-TMs (TM = Fe, Co, Ni, Cu, Zn) possessing the same structure as that of the 2D MOF-Co we reported earlier, based on density functional theory (DFT) calculations.

In this study, we first investigated the adsorption behaviors of $N_2$ and $CO_2$ molecules towards the surfaces of 2D MOFs. By analyzing the adsorption energy, charge transfer, and corresponding bond lengths of the adsorbates, the interaction between adsorbates and 2D MOFs was clarified. The electrocatalytic NRR and $CO_2$RR pathways for those chemisorbed molecules on the catalyst surfaces were investigated. Our theoretical results indicate that 2D MOF-Fe is an excellent bi-functional electrocatalyst for both NRRs and $CO_2$RRs. Moreover, 2D MOF-Co and MOF-Ni exhibit high performance only for electrocatalytic NRRs, while the other 2D MOF-TM materials are not appropriate catalysts for electrocatalytic NRRs or $CO_2$RRs.

## 2. Calculation Methods

All calculations were performed in the DFT framework using the Vienna ab initio simulation package (VASP) [39,40]. Generalized gradient approximation (GGA) under the projector-augmented plane wave (PAW) method was used [41–43]. The exchange correlation functions were set in the form of Perdew–Burke–Ernzerhof (PBE) [44,45]. In addition, the DFT-D2 correction method of Grimme was applied to describe van der Waals (vdW) forces [46,47]. The previously prepared 2D MOF-Co possesses a defined geometric structure and its unit cell has the chemical formula of $C_{16}H_{10}O_{10}Co_3$, with triclinic symmetry (space group P$\bar{1}$) [34]. To model other 2D MOF-TMs (TM = Fe, Ni, Cu, Zn), the Co atoms in the 2D MOF-Co structure were replaced by TM atoms. In this study, 2D MOF-TM catalysts were constructed with the p(2 × 2) supercell and with a sufficient vacuum region (20 Å) perpendicular to the surface, as shown in Figure 1. To simulate those structures and the corresponding adsorption configurations, the plane wave

truncation energy used was 400 eV, and the energy convergence criterion was set to $10^{-4}$ eV. The convergence threshold of $10^{-2}$ eV/Å was applied for force. The Monkhorst-Pack of $3 \times 2 \times 1$ K-points were used for the Brillouin-zone integration. We completed a Bader charge analysis to investigate the oxidation state of the mental center. Calculation results indicated that the oxidation state for the tetra- and six-coordinated metal atoms (the tera- and six-coordinated metal atoms were marked as i and ii by dotted blue circles in Figure 1) were different. The oxidation states of atom-i/ii were 1.15/1.31 for Fe, 1.04/1.28 for Co, 1.02/1.23 for Ni, 1.04/1.20 for Cu, and 1.31/1.39 for Zn.

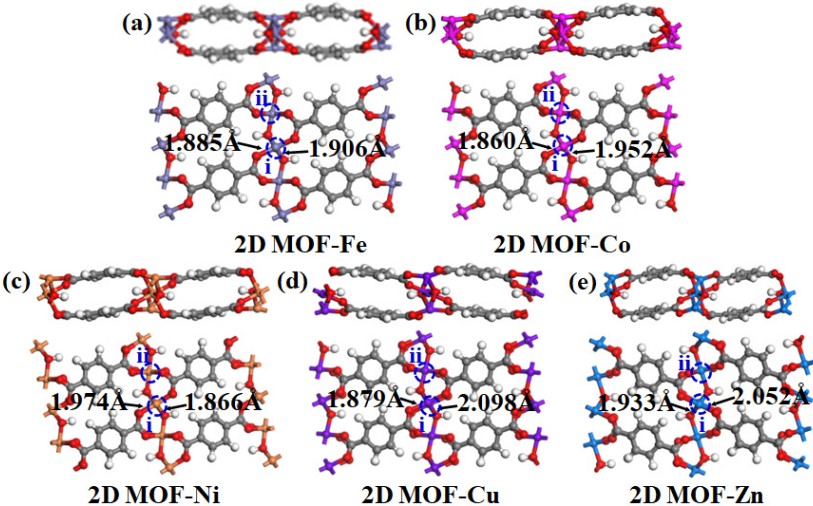

**Figure 1.** Side and top views of geometric structures for (**a**) MOF-Fe, (**b**) MOF-Co, (**c**) MOF-Ni, (**d**) MOF-Cu, and (**e**) MOF-Zn. Color cards: C: Grey, H: White, O: Red, Fe: Lilac, Co: Pink, Ni: Orange, Cu: Purple, Zn: Sky blue.

To investigate the adsorption of $N_2$ and $CO_2$ towards the 2D MOF-TM surface, the isolated $N_2$ and $CO_2$ molecules were previously simulated in a large cubic cell of 15 Å in length. The adsorption energies were defined as $E_{ads} = E_{MOF-gas} - E_{molecule} - E_{MOF}$, where $E_{MOF-gas}$, $E_{molecule}$, and $E_{MOF}$ are the total energies of the 2D MOF-TM surface with adsorbed $N_2$ or $CO_2$ molecules, the isolated $N_2$ or $CO_2$ molecule and the clean 2D MOF-TM surface, respectively. A Bader charge analysis and difference charge density calculations were carried out to clarify the interaction between adsorbed molecules and the catalyst surfaces. The electrocatalytic NRR and $CO_2$RR both involve several coupled proton and electron transfers. To acquire the free energy profiles of electrocatalytic $N_2$ and $CO_2$ reduction reactions, the computational electrode model (CHE) was employed [48,49]. According to the CHE, the energy changes are for reactions with the proton–electron pairs $(H^+ + e^-)$. The chemical potential of $(H^+ + e^-)$ can be obtained through the equation of $\mu(H^+ + e^-) = 1/2\mu(H_2)$-eU, where $\mu(H_2)$ is the free energy of gaseous $H_2$ and U is the applied bias. The free energy change ($\Delta G$) at each elementary step of the NRR and $CO_2$RR processes was calculated using the equation $\Delta G = \Delta E$-neU $+ \Delta ZPE - T\Delta S$, where $\Delta E$ is the change in electron energy calculated by DFT, n is the number of transferred electrons involved in the elementary reaction, $\Delta ZPE$ is the zero-point energy difference, T is the reaction temperature (in this paper, all are at room temperature, T = 298.15 K), $\Delta S$ is the change in entropy value. $\Delta ZPE$ and $\Delta S$ can be obtained through a frequency analysis.

## 3. Results and Discussion

### 3.1. Adsorption Behaviors of $N_2$ and $CO_2$ on the 2D MOF-TM Surface

In order to evaluate whether MOF-TMs can be used as electrocatalysts for NRR and $CO_2$RR, the adsorption behaviors of $N_2$ molecules on the 2D MOF-TM surfaces were systematically investigated firstly. Various molecule adsorption configurations for $N_2$

towards different sites on the catalyst surfaces were taken into consideration to determine their stable adsorption geometries.

Calculation results show that the adsorption states of $N_2$ on the MOF-Cu and MOF-Zn surfaces are physically adsorbing, with adsorption energy of −0.11 eV and −0.10 eV, respectively (Figure S1). The physical adsorption modes suggest that the 2D MOF-Cu and MOF-Zn designed in this work are not suitable electrocatalysts for NRRs, while $N_2$ can be chemisorbed on the other three MOF-TM (TM = Fe, Co, Ni) surfaces at metal sites in end-on configuration (Figure 2a–c). Table 1 summarizes the adsorption parameters for $N_2$ chemical adsorption ($N_2$*) towards those three 2D MOF-TM materials. The corresponding adsorption energies of $N_2$* were calculated to be −1.43 eV, −1.11 eV, and −0.64 eV for MOF-Fe, MOF-Co, and MOF-Ni, respectively. The N-N bond lengths for $N_2$* on MOF-Fe, MOF-Co, and MOF-Ni, respectively, are 1.139 Å, 1.135 Å, and 1.126 Å, all greater than that of the isolated $N_2$ molecule (1.117 Å). In addition, newly formed TM-N bonds with bond lengths of 1.773 Å (Fe-N), 1.758 Å (Co-N), and 1.812 Å (Ni-N) were observed. Further difference charge density calculations and Bader charge analyses were carried out to further declare the interaction between $N_2$* and MOF-Fe/Co/Ni. The difference charge density plots presented in Figure 2 denote that obvious charge redistribution occurs due to the adsorption interaction [49,50]. According to the Bader charge analysis, electrons are transferred from the MOF-TM surface to $N_2$* and the net obtained charge is 0.27 e, 0.21 e, and 0.09 e for $N_2$* on MOF-Fe, MOF-Co, and MOF-Ni, respectively. Moreover, the PDOS plots in Figure 3 show that the PDOS peaks for the two N atoms of $N_2$* are broadly dispersed in comparison with those of the isolated $N_2$ molecule [51,52]. And, overlap between the $N_1$ atom and metal atom occurs in the spectrum, especially in the energy ranges of −6.0 eV to −8.0 eV and 2.0 eV to 3.0 eV. According to frequency calculations, the red shifts of the N-N stretching vibrational frequencies for $N_2$ bound to the Fe, Co, and Ni 2D MOF were 141 $cm^{-1}$, 103 $cm^{-1}$, and 23 $cm^{-1}$, respectively. And, the changes in the N-N stretch frequencies could be used as good indicators for metal-$N_2$ binding interaction. These above results signify that $N_2$ molecules can be activated effectively by 2D MOF-Fe/Co/Ni.

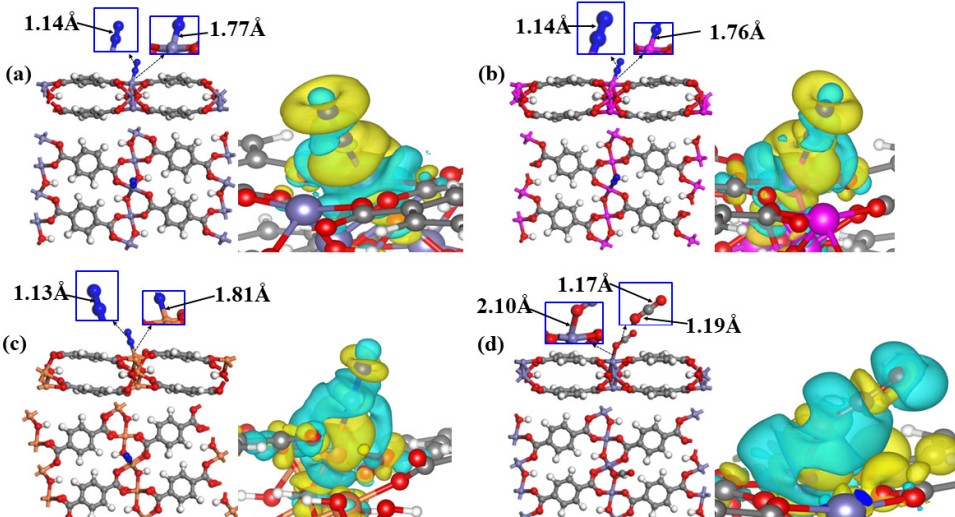

**Figure 2.** Side and top views of adsorption configurations and the corresponding difference charge density plots for (**a**) $N_2$ adsorption on MOF-Fe, (**b**) $N_2$ adsorption on MOF-Co, (**c**) $N_2$ adsorption on MOF-Ni, (**d**) $CO_2$ adsorption on MOF-Fe. Color cards: C: Gray, H: White, O: Red, Fe: Lilac, Co: Pink, Ni: Orange, Cu: Purple, Zn: Sky blue. In different charge density plots, the blue (yellow) wireframes denote the loss (gain) of electrons with the isosurface values set as 0.003 $Å^{-3}$.

**Table 1.** Chemical adsorption parameters for $N_2^*$ and $CO_2^*$ on 2D MOF-TM surfaces: adsorption energy ($E_{ads}$), N-N bond length of $N_2^*$ ($L_{N-N}$), C-O bond length of $CO_2^*$ ($L_{C-O}$), the distance between TM sites and N atom of $N_2^*$ ($d_{TM-N}$), the distance between Fe site and the O atom of $CO_2^*$ ($d_{Fe-O}$), the value of net transferred from MOF-TM to $N_2^*$ and to $CO_2^*$ ($\Delta q$), and the redshift values of N-N stretch frequencies with respect to the free $N_2$ molecule ($\Delta k$).

| $N_2^*$ on MOF-TM | $E_{ads}$ (eV) | $L_{N-N}$ (Å) | $d_{TM-N}$ (Å) | $\Delta q$ | $\Delta k$ (cm$^{-1}$) |
|---|---|---|---|---|---|
| Fe | −1.43 | 1.139 | 1.773 | 0.27 | 141 |
| Co | −1.11 | 1.135 | 1.758 | 0.21 | 103 |
| Ni | −0.64 | 1.126 | 1.812 | 0.09 | 23 |
| $CO_2^*$ on MOF-Fe | −0.44 | $L_{C-O}$ (Å) | $d_{Fe-O}$ (Å) | 0.01 | |
| | | 1.185/1.172 | 2.100 | | |

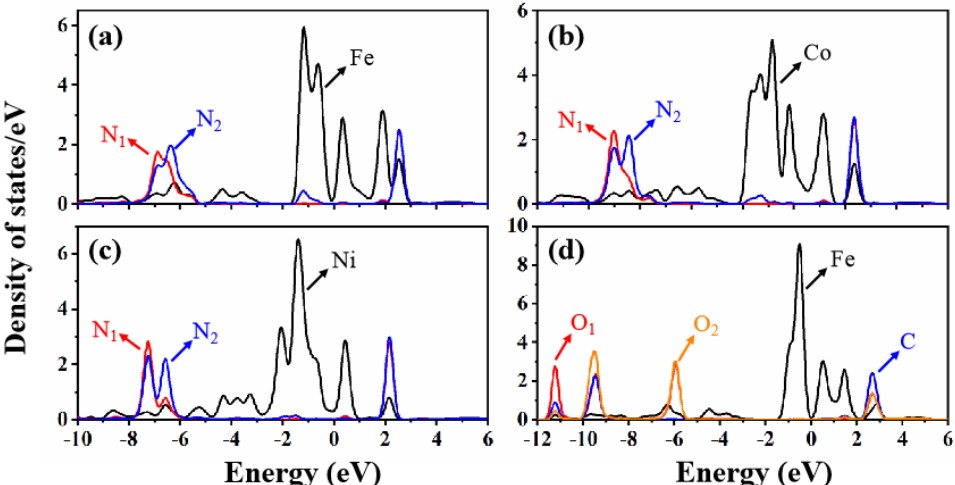

**Figure 3.** PDOS plots for (**a**) $N_2$ adsorption on MOF-Fe, (**b**) $N_2$ adsorption on MOF-Co, (**c**) $N_2$ adsorption on MOF-Ni, (**d**) $CO_2$ adsorption on MOF-Fe. The Fermi level was assigned at 0 eV. $N_1/O_1$ represents the N/O atom bonded with surface metal site, and $N_2/O_2$ is the N/O atom located far from the metal site.

For $CO_2$ adsorption on the 2D MOF-TM, various initial adsorption modes were also examined. Full optimization of the initial structures reveals the physical adsorption of $CO_2$ on MOF-Co, Ni, Cu, and Zn, with a small adsorption energy of −0.21 eV, −0.21 eV, −0.19 eV, and −0.17 eV and with remote distances between the $CO_2^*$ and MOF-TM surfaces, as shown in Figure S2. Therefore, those four 2D MOF-TM materials should not be efficient electrocatalysts for $CO_2$RR. Fortunately, adsorption of $CO_2$ on the 2D MOF-Fe led to a chemical adsorption state with $CO_2^*$ in the linear structure, where electron redistribution between $CO_2^*$ and the MOF-Fe surface, and a newly formed bond between the O atom of $CO_2^*$ and the Fe site was observed (Figure 2d). The corresponding adsorption parameters for $CO_2$ on the MOF-Fe were also listed in Table 1. We can see that the adsorption energy was determined to be −0.44 eV and the newly formed O-Fe bond was 2.100 Å. Difference charge density calculations and Bader charge analyses show that the net charge transferred from MOF-Fe to $CO_2^*$ was 0.01 e. Similar with $N_2^*$, the PDOS peaks for the C and O atoms of $CO_2^*$ are more dispersed compared to those for the isolated $CO_2$ molecule. And, an obvious overlap between the $O_1$ atom and the surface Fe atom can be seen in Figure 3d. Significantly, the $CO_2$ was polarized and activated weakly by the 2D MOF-Fe surface, reflected by the change in C-O bond lengths (1.185 Å and 1.172 Å) in $CO_2^*$ relative to those in isolated $CO_2$ (1.177 Å), and a bent angle of ∠O-C-O = 177.5° for the adsorbed $CO_2^*$.

*3.2. Electrocatalytic Processes and Mechanisms for NRR and CO$_2$RR*

Previous reports show that the chemisorption of N$_2$ and CO$_2$ on the catalyst surface is favorable for the electrocatalytic N$_2$ and CO$_2$ reduction reaction. Therefore, we herein focused on the electrocatalytic NRRs on the 2D MOF-Fe, MOF-Co, and MOF-Ni, and CO$_2$RRs on the 2D MOF-Fe, according to the above adsorption property calculations.

Figure 4 shows the corresponding free energy changes for each hydrogenation step of the NRR process on the 2D MOF-Fe surface at a zero electrode potential (U = 0 V). The 'solvation corrections' by the COSMO solvation model were obtained with H$_2$O as the solvent. In the first step of the reaction process, NN* is hydrogenated to NNH* by a coupling reaction with H$^+$ and e$^-$ pair. In that step, the N-N bond is extended from 1.139 Å to 1.213 Å, and the value of ΔG for this process is 0.84 eV. In the second hydrogenation step, both NNH$_2$* and NHNH* formations are slightly upslope, with a ΔG of 0.05 eV and 0.03 eV, respectively. The almost identical energy between NNH$_2$* and NHNH* denotes that alternating and distal pathways happened simultaneously for the first two hydrogenation steps. Subsequently, formations of NH$_2$NH* and NH$_3$* are endothermic processes, while NHNH$_2$* formation is exothermic with a ΔG of −0.35 eV and −0.37 eV. Therefore, the third hydrogenation step leads to NHNH$_2$*. Then, NH$_2$NH$_2$* rather than NH* + NH$_3$ is gained by the addition of the fourth H, and the corresponding ΔG value was −0.31 eV. For the subsequent two reaction steps to produce NH$_2$* + NH$_3$(g) and NH$_3$* + NH$_3$(g), the corresponding ΔG values were −1.24 eV and −0.64 eV, respectively. In the final hydrogenation step, the Fe-N bond length is extended to 1.997 Å, which facilitates the desorption of NH$_3$ due to the weakened Fe-N bond. Moreover, though the free energy gain for the release of the second NH$_3$ molecule was calculated to be as large as 1.11 eV, NH$_3$ release, which can be facilitated by the high solubility of NH$_3$ in water, is not considered as an elementary reaction of the NRR [52]. According to the above results, the overall NRR process happened along the alternating path and the first hydrogenation step was determined as the potential-limiting step with a free energy increase of 0.84 eV. The free energy increase in the potential-limiting step obtained without solvent corrections is 0.88 eV, almost uniform with the value with solvent corrections, indicating that the solvent effect is weak for NRRs on 2D MOF-metals.

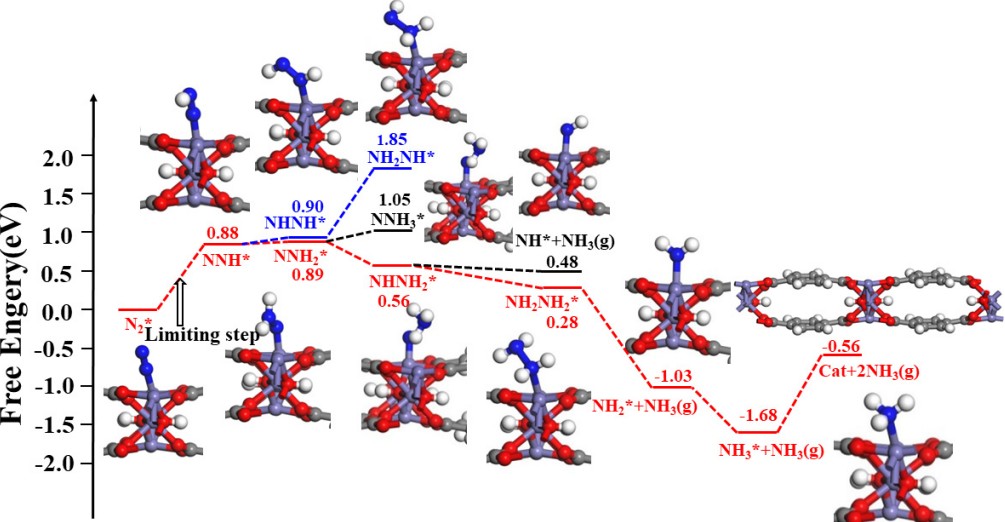

**Figure 4.** Free energy diagram of the NRR process of N$_2$* on 2D MOF-Fe and the corresponding geometries of intermediates and products. White, blue, lilac, grey, and red balls denote H, N, Fe, C, and O atoms, respectively.

The structures of the reaction intermediates and the corresponding Gibbs free energy profiles for NRRs on the 2D MOF-Co are shown in Figure 5. The first hydrogenation step results in N-N bonds extended from 1.135 Å to 1.204 Å and the value of ΔG is 1.00 eV. In the second step, NHNH* formation has an advantage over NNH$_2$* because the energy of

NHNH* was 0.30 eV lower that of NNH₂* and NHNH* formation from NNH* is downhill, with a ΔG of −0.25 eV. In the third step, the formation of *NHNH₂ is exothermic, at ΔG −0.16 eV. All the subsequent three hydrogenation steps are downhill in free energy, leading to intermediates of NH₂NH₂*, NH₂* + NH₃(g), and NH₃* + NH₃(g), and the corresponding ΔG value is −0.51 eV, −1.04 eV, and −0.95 eV, respectively. The bond length of Co-NH₃* is extended to 1.957 Å and the second NH₃ release needs to overcome the 1.02 eV energy barrier. Overall, the NRR process was along the alternating pathway and the potential-limiting step was determined at the first hydrogenation step, with an energy barrier of 1.00 eV.

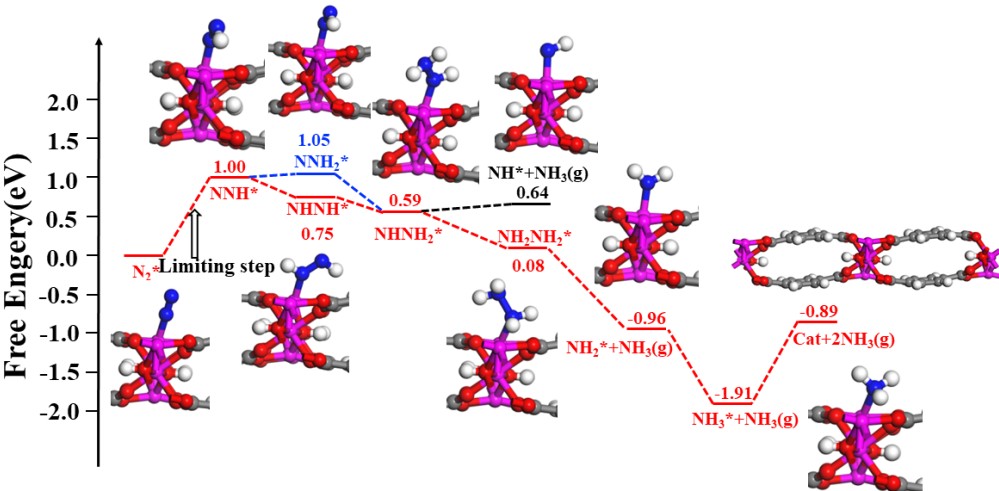

**Figure 5.** Free energy diagram of the NRR process of N₂* on 2D MOF-Co and the corresponding geometries of intermediates and products. White, blue, pink, grey, and red balls denote H, N, Co, C, and O atoms, respectively.

The electrocatalytic NRR pathways on the 2D MOF-Ni are shown in Figure 6. Similar to the previous two catalytic NRR processes on the 2D MOF-Fe and MOF-Co, the first hydrogenation step is determined as the potential-limiting step and the corresponding energy gain is 1.17 eV. In the same way, the N-N bond is increased from 1.126 Å to 1.201 Å. Note that the first three hydrogenation steps are along the alternating pathway, while the fourth hydrogenation step takes place with a distal mechanism, reflected by the fact that the free energy of NHNH₃* was 0.39 eV lower than that of NH₂NH₂*. Then, the fifth and the sixth hydrogenation steps are along the alternating pathway again. Therefore, the NRR process on the 2D MOF-Ni is the mix of alternating and distal pathways. The ΔG values for the second to the sixth hydrogenation step are −0.68 eV, −0.05 eV, −1.26 eV, −0.80 eV, and −1.76 eV, respectively. Moreover, the bond length between the Ni site and the N atom of the second NH₃* is 1.909 Å, much shorter than those between the Fe/Co site and the N atom of NH₃*. Therefore, the desorption energy of the second NH₃ release from the Ni site is 2.43 eV, larger than those values for NH₃ release from the Fe/Co site (1.12/1.02 eV).

The entire electrocatalytic NRR processes of 2D MOF-Fe, MOF-Co, and MOF-Ni were systematically examined. According to the above results, the energy increase in the potential-limiting step for NRRs on the 2D MOF-Fe (0.88 eV), MOF-Co (1.00 eV) and MOF-Ni (1.17 eV) is less than or at least comparable with those reported values of many efficient electrocatalysts for NRRs (0.85 eV on FeN₄ site of FePc [53] 0.85 eV on Nb₃c of SnN₂O₆ nanosheet [54] 1.23 eV on N-C@NiO/GP catalyst [55] 1.47 eV on WO₃-x nanosheet [56]), meaning that the three 2D MOF-TM materials hold immense potential to be used as electrocatalysts for NRRs.

Here we discuss the investigation of electrocatalytic CO₂RRs on the 2D MOF-Fe. The solvent effect is taken into consideration herein. The CO₂ hydrogenation pathway and free energy profiles over the MOF-Fe catalyst are shown in Figure 7. In the hydrogenation process, both the two O sites and C sites of each intermediate were considered for H

addition. For the first H addition, OCHO* intermediate instead of OCOH* and OH* + CO* is formed and the free energy gain ΔG was −0.68 eV. The negative value of ΔG signified the easy obtaining of OCHO*. For further hydrogenation, O* + $CH_2O$ and OHCHO* formation, respectively, require ΔG of 1.37 eV and 0.22 eV, while the OCHOH* formation reaction is exothermic with a ΔG of −0.13 eV, indicating that OCHOH* is favorable to form, followed by the hydrogenation of OCHOH* to $OCH_2OH*$ with a free energy uphill and the ΔG was calculated to be 0.16 eV. Then, the fourth H would attack the O site of $OCH_2OH*$, which is orientated to the surface, resulting in $OHCH_2OH*$ with a ΔG of −0.25 eV. The followed $CH_3OH$ formation from the $OHCH_2OH*$ hydrogenation is determined as the potential-limiting step with a ΔG of 0.84 eV. Excitingly, $CH_3OH$ is exclusively the carbon products. The ΔG value of 0.84 eV was lower than those values with a range of 0.9~1.1 eV on Cu-based materials, which were known as excellent catalysts for $CO_2$ electroreduction to hydrocarbons and oxygenates [57,58]. The remaining OH* reacts with the sixth H to form a $H_2O*$ with a ΔG of −0.25 eV. Finally, the $H_2O$ molecule peels off from the catalyst with a desorption energy of −0.20 eV and the surface Fe sites are reactivated. These results indicate that the 2D MOF-Fe is an appropriate electrocatalyst for $CO_2RRs$ with a high efficiency and high product selectivity.

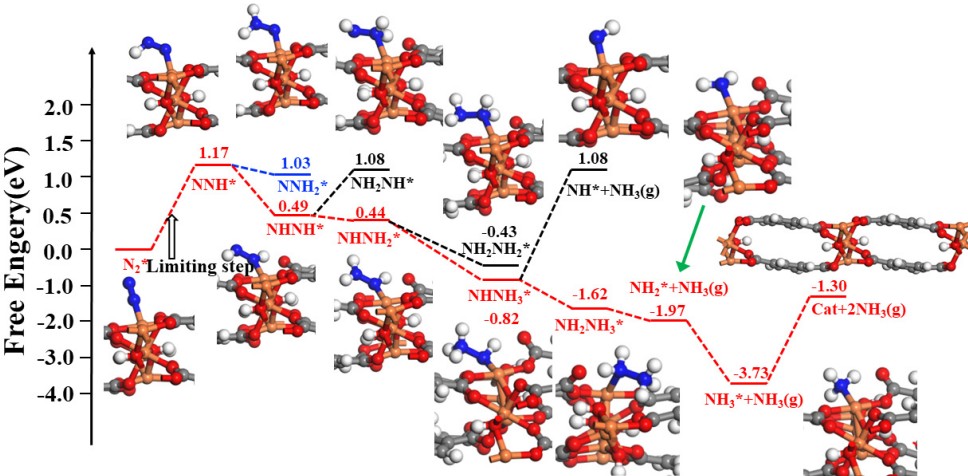

**Figure 6.** Free energy diagram of the NRR process of $N_2*$ on 2D MOF-Ni and the corresponding geometries of intermediates and products. White, blue, orange, grey, and red balls denote H, N, Ni, C, and O atoms, respectively.

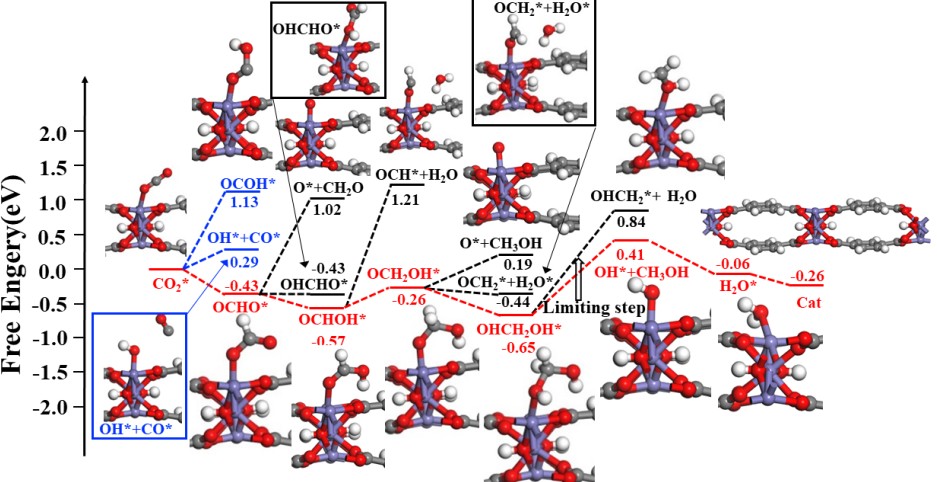

**Figure 7.** Free energy diagram of the $CO_2RR$ process of $CO_2*$ on 2D MOF-Fe and the corresponding geometries of intermediates and products. White, lilac, grey, and red balls denote H, Fe, C, and O atoms, respectively.

## 4. Conclusions

In brief, we have systematically investigated the application prospects of five 2D MOF-TMs (TM = Fe, Co, Ni, Cu, Zn) as electrocatalysts for NRRs and $CO_2$RRs by DFT calculations. The results show that $N_2$ molecules can be effectively activated by the surface metal site of the 2D MOF-Fe, MOF-Co, and MOF-Ni due to the chemisorption interaction. The appropriate free energy increase in the potential-limiting step (0.84 eV for NRRs on MOF-Fe, 1.00 eV for NRRs on MOF-Co, and 1.17 eV for NRRs on MOF-Ni) signifies that those three 2D MOF-TM materials hold promising applications in NRRs. Furthermore, $CO_2$ molecules could be selectively reduced to a single hydrocarbon product, $CH_3OH$, on a 2D MOF-Fe catalyst through the pathway $CO_2^* \rightarrow OCHO^* \rightarrow OCHOH^* \rightarrow OCH_2OH^* \rightarrow OHCH_2OH^* \rightarrow CH_3OH$. And, the fifth hydrogenation step was determined as the potential-limiting step, with a free energy increase of 0.84 eV. These calculation results provide viable approaches to further design and screen the electrocatalysts for NRRs or $CO_2$RRs.

**Supplementary Materials:** The following supporting information can be downloaded at: https://www.mdpi.com/article/10.3390/cryst13101426/s1. Figure S1. Side and top views of adsorption configurations for $N_2$ on (a) MOF-Cu, (b) MOF-Zn. Color cards: C: Grey, H: White, O: Red, Cu: Purple, Zn: Sky blue. Figure S2. Side and top views of adsorption configurations for $CO_2$ on (a) MOF-Co, (b) MOF-Ni, (c) MOF-Cu, (d) MOF-Zn. Color cards: C: Grey, H: White, O: Red, Co: Pink, Ni: Orange, Cu: Purple, Zn: Sky blue.

**Author Contributions:** A.S.: Proposed the research idea, Designed the research plan, First draft of the paper. M.W.: Conducted the experiment, Edited the paper. S.L.: Data statistics. Y.D.: Supervision, Project administration, Funding support. D.W.: Study method design, Final version revision, Project administration, Funding support. All authors have read and agreed to the published version of the manuscript.

**Funding:** This research was funded by National Natural Science Foundation of China (21978110, 51772126, 21801092, and 11904129), the Program for the Development of Science and Technology of Jilin Province (20210101409JC, YDZJ202201ZYTS307, 20200201187JC, 20200801040GH), University-Industry Collaborative Education Program (202002284033). Research on constructing the training system of innovative talents in teachers' major in digital era (JS2360).

**Acknowledgments:** Computing time granted by the Computing Center of Jilin Province is acknowledged.

**Conflicts of Interest:** The authors declare no conflict of interest.

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
