# Peer review of "Two-Dimensional Metal–Organic Framework TM Catalysts for Electrocatalytic N2 and CO2 Reduction: A Density Functional Theory Investigation"

_crystals, doi:10.3390/cryst13101426_

Round 1

Reviewer 1 Report

The study presented by the authors investigates the potential of two-dimensional metal-organic framework (MOF) materials as efficient electrocatalysts for the reduction of N2 and CO2 using density functional theory (DFT) calculations. The research is well-conducted, and the findings are promising. However, there are some points that need revisions before publication. I think that this work can be accepted in this journal after some revisions that may help to improve this version:

1. The authors calculate the Fe/Co/Ni elements in this paper. I am curious about whether the authors take the influence of spin multiplicity as consideration. The Fe/Co/Ni atoms tend to have high spin multiplicity or low spin state?

2. Have the authors considered the choice of reaction sites, there may be multiple reaction sites due to the complexity of MOF materials.

3. I suggest that the coordinate files of some key models should be provided in SI.

4. The authors should highlight the key energy steps in the free energy reaction path diagram.

5. The methods used in the article, such as the difference charge density, DOS orbital analysis, have been widely used in many other works describing the adsorption mechanism, which could be included in article with refs Adv. Mater. (10.1002/adma.202305074), Nanoscale (10.1039/d2nr06665c), Small (10.1002/smll.202206750).

Minor editing of English language required

Author Response

Comment from Reviewer #1: Comments and Suggestions for Authors

The study presented by the authors investigates the potential of two-dimensional metal-organic framework (MOF) materials as efficient electrocatalysts for the reduction of N2 and CO2 using density functional theory (DFT) calculations. The research is well-conducted, and the findings are promising. However, there are some points that need revisions before publication. I think that this work can be accepted in this journal after some revisions that may help to improve this version:

Common 1: The authors calculate the Fe/Co/Ni elements in this paper. I am curious about whether the authors take the influence of spin multiplicity as consideration. The Fe/Co/Ni atoms tend to have high spin multiplicity or low spin state?

Response 1: Thanks for your suggestion. We don’t consider the effect of Fe/Co/Ni atoms spin multiplicity. First of all, the calculation of spin multiplicity greatly affects the calculation speed, and the requirements for the equipment are relatively high. Second, according to our previous experience, whether or not to increase the spin calculation doesn’t affect the energy gain for each step in NRR path. Therefore, spin multiplicity isn’t taken into account in the calculation.

Common 2: Have the authors considered the choice of reaction sites, there may be multiple reaction sites due to the complexity of MOF materials.

Response 2: Thanks for your suggestion. In fact, we considered a variety of situations in the initial N2 adsorption, including the end-on and side-on adsorption configurations on different reaction sites. After careful consideration of the energy of N2 adsorption on different reaction sites, N2 adsorption on Ni site was determined as the most stable adsorption structure.

Common 3: I suggest that the coordinate files of some key models should be provided in SI.

Response 3: Thanks for your suggestion. According to your suggestion, we have placed some key models information in “Supplementary Materials”.

Common 4: The authors should highlight the key energy steps in the free energy reaction path diagram.

Response 4: Thanks for your suggestion. According to your suggestion, we have modified Figure 4.5.6.7 to highlight key energy steps in the free energy reaction path diagram. And replace the modified image with the original image.

Common 5: The methods used in the article, such as the difference charge density, DOS orbital analysis, have been widely used in many other works describing the adsorption mechanism, which could be included in article with refs Adv. Mater. (10.1002/adma.202305074), Nanoscale (10.1039/d2nr06665c), Small (10.1002/smll.202206750).

Response 5: Thanks for your suggestions. According to your suggestion, the above three literatures were cited as Ref 49-51 in the main text and added in the Reference list. Correspondingly, the reference serial number was adjusted.

Reviewer 2 Report

Please, find attached my comments.

Author Response

Comment from Reviewer #2:

This is a good work, however the following revisions should be taken into consideration:

Common 1: Literature should be updated.

Response 1: Thanks for your suggestion. According to your suggestion, we have added three more recent articles and adjusted the sequence number of other references.

Common 2: The number of the last Figure’s caption should be corrected. It is not Figure 1, as has been denoted by the authors.

Response 2: Thanks for your suggestion. According to your suggestion, we have modified the number of the last Figure’s caption.

Common 3: In literature is referred that the parallel and vertical N2 molecule adsorption shows completely different reactivity and selectivity. Have the authors studied the various way of N2 adsorption?

Response 3: Thanks for your suggestion. In fact, we considered a variety of situations in the initial design, respectively considering the combination of N2 and Ni atoms, N2 and Ni atoms combined with O atoms, and N2 combined with three C atoms. Finally, it is determined that the energy of N2 and Ni atoms is the lowest. Therefore, the adsorption site of N2 is determined on the Ni atoms.

Common 4: Fig.2 is not so clear, especially at the points that that arrows indicate. Furthermore, it is suggested a magnifying picture at the points that the arrows indicate, to be presented.

Response 4: Thanks for your suggestion. According to your suggestion, we have modified the points indicated by the arrows in Figure 2 and attached the enlarged image.

Common 5: Did the authors take into consideration the angle of adsorption of the molecules?

Response 5: Thanks for your suggestion. In fact, different adsorption angles are taken into account when MOF adsorbs N2 or when N2 and H2 are combined. Finally, the adsorption angle with the lowest energy is determined for the next step.